# Neural Graph Control Barrier Functions Guided Distributed Collision-avoidance Multi-agent Control

**Songyuan Zhang**
Department of Aeronautics and Astronautics
Massachusetts Institute of Technology
`szhang21@mit.edu`

**Kunal Garg**
Department of Aeronautics and Astronautics
Massachusetts Institute of Technology
`kgarg@mit.edu`

**Chuchu Fan**
Department of Aeronautics and Astronautics
Massachusetts Institute of Technology
`chuchu@mit.edu`

**Abstract:** We consider the problem of designing distributed collision-avoidance multi-agent control in large-scale environments with potentially moving obstacles, where a large number of agents are required to maintain safety using only local information and reach their goals. This paper addresses the problem of collision avoidance, scalability, and generalizability by introducing graph control barrier functions (GCBFs) for distributed control. The newly introduced GCBF is based on the well-established CBF theory for safety guarantees but utilizes a graph structure for scalable and generalizable decentralized control. We use graph neural networks to learn both neural a GCBF certificate and distributed control. We also extend the framework from handling state-based models to directly taking point clouds from LiDAR for more practical robotics settings. We demonstrated the efficacy of GCBF in a variety of numerical experiments, where the number, density, and traveling distance of agents, as well as the number of unseen and uncontrolled obstacles increase. Empirical results show that GCBF outperforms leading methods such as MAPPO and multi-agent distributed CBF (MDCBF). Trained with only 16 agents, GCBF can achieve up to 3 times improvement of success rate (agents reach goals and never encountered in any collisions) on $< 500$ agents, and still maintain more than $50\%$ success rates for $> 1000$ agents when other methods completely fail.[1]

**Keywords:** Distributed control, Control barrier functions, Graph neural networks

## 1 Introduction

Multi-agent systems (MAS) can complete much more complex tasks efficiently as compared to single-agent systems such as reconnaissance or sensor coverage of a large unexplored area. Safety of MAS, in terms of collision and obstacle avoidance, is a non-negotiable requirement in the numerous autonomous robotics applications (see [1] for an overview) such as a swarm of drones flying in a dense forest [2, 3], multi-object configuration and manipulation in warehouses [4, 5, 6] and autonomous driving [7, 8, 9]. In addition, the agents are required to either follow a pre-defined path or reach a destination for completing their individual or team objectives. With the increase in the number of robots in the MAS, it becomes difficult to design control policies for all the agents for such a multi-task problem as the computational complexity grows exponentially with the MAS scale [10].

Common multi-agent motion planning methods include solving mixed integer linear programs (MILP) for computing safe paths for agents [11, 12] and RRT-based methods [13]. However, they are not

---

[1]Project website: https://mit-realm.github.io/gcbf-website/

7th Conference on Robot Learning (CoRL 2023), Atlanta, USA.

scalable to large-scale MAS. Multi-agent Reinforcement Learning (MARL) approaches, e.g., Multi-agent Proximal Policy Optimization (MAPPO) [14], have also been adapted to solve the problems. However, most of the MARL works model safety as a penalty rather than a hard constraint and thus, cannot guarantee safety. In recent years, safety constraints have been handled via control barrier functions (CBFs) [15]. Particular for MAS, generally a CBF is assigned for each safety constraint, and then an approximation method is used for accounting for the multiple constraints [16, 17, 18, 19]. Their issue is that it is very difficult to construct a handcrafted CBF for large-scale MAS consisting of highly nonlinear dynamics. The Multi-agent Decentralized CBF (MDCBF) framework in [20] uses a neural network-based CBF designed for MAS, but they do not encode a method of distinguishing between other controlled agents and *uncontrolled* agents such as static and dynamic obstacles. This can lead to either conservative behavior if all the neighbors are treated as non-cooperative obstacles, or collisions if the obstacles are treated as cooperative controlled agents. Furthermore, they use a discrete approximation of the time derivative of the CBF but do not account for changing graph topology, which can lead to a wrong evaluation of the CBF constraints and consequently, failure. We provide further empirical evidence of a significant drop in the performance of MDCBF as the number of agents increases through various numerical experiments. The Control Admissiblity Models (CAM)-based framework in [21] also attempts to address a similar problem. However, it involves sampling control actions from a set defined by CAM and cannot always find a feasible control input.

To overcome these limitations, in this paper, we present a novel Graph CBF for large-scale MAS to address the problem of safety, scalability, and generalizability. We propose a learning-based control policy to achieve a higher safety rate in practice. We use graph neural networks (GNN) to better capture the changing graphical topology of distance-based inter-agent communications. We also use LiDAR-based observations for handling unseen and potentially nonstationary obstacles in real-world environments. With these technologies, our proposed framework can generalize well to many challenging settings, including more crowded environments and unseen obstacles.

We consider two 2D and one 3D environment in our experiments. In the obstacle-free case, we train with 16 agents and test with over 1000 agents. For $< 500$ agents, our method achieves a threefold improvement in safely reaching tasks, while for large-scale experiments ($> 1000$ agents) where the baselines achieve close to 0 success rate, our approach achieves $50\% - 100\%$ success rate. In the obstacle environment, we consider only 16 point-sized obstacles in training, while in testing, we consider up to 32 large obstacles. We see over $15\%$ improvement in success rate over baselines. The experiments corroborate that the proposed method outperforms the existing methods in successfully completing the tasks in various 2D and 3D environments. Our contributions are summarized below:

- We introduce Graph CBF (GCBF), a new kind of barrier function for MAS to encode collision avoidance constraints and to handle different types of agents and obstacles.

- We use GNNs to jointly learn a GCBF and a distributed controller which is robust to the changes of neighbors, and a LiDAR-based observation model for obstacles.

- Empirical performance shows a significant improvement by our GCBF over other leading approaches, especially in difficult settings.

**Related work** Graph-based planning approaches such as *prioritized* multi-agent path finding [22], conflict-based search for MAPF [23] can be used for multi-agent path planning for known environments. However, these offline path-planning methods cannot be used for dynamic environments or large-scale systems due to computational complexity. Another line of work for motion planning in obstacle environments is based on the notion of velocity obstacles [24, 25] defined using collision cones for velocity. Such methods can be used for large-scale systems with safety guarantees under mild assumptions. However, the current frameworks under this notion assume single or double integrator dynamics for agents. The work in [26] scales to large-scale systems, but it only considers discrete action space and hence does not apply to robotic platforms that use more general continuous input signals. Works such as [27, 28, 29] address this problem using GNNs for generalization to unseen environments and are shown to work on teams of up to a hundred agents. However, they are not scalable to very large-scale problems (e.g., a team of 1000 agents) due to the computational bottleneck.

In recent years, the most commonly employed method of solving safe motion planning problems involves neural CBF-based approaches [20, 30, 31, 32]. Machine learning (ML)-based approaches have shown promising results in designing CBF-based controllers for complex safety-critical systems [30, 31, 32]. The NN-CBF framework consists of model-based learning [33, 32, 34, 35] or model-free learning [36, 37, 38]. Our approach uses a model-based learning framework, and in contrast to the aforementioned works, applies to MAS. Utilizing the permutation-invariance property, GNN-based methods have been employed for problems involving MAS [28, 27, 39, 21, 40, 41, 42, 29, 43]. These prior work only consider static obstacles, or do not consider the presence of obstacles or *uncontrolled* agents at all. On the other hand, there is also a lot of work on MARL-based approaches with focuses on motion planning [44, 45, 46, 47, 48, 49, 14, 50, 51, 52]. But these approaches cannot provide safety guarantees due to the reward structure and as argued in [53], MARL-based methods are still in the initial phase of development when it comes to safe multi-agent motion planning.

## 2 Problem formulation

We consider the problem of designing a distributed control framework to drive $N$ agents $V_a := \{1, 2, \ldots, N\}$ to their goal locations while avoiding collisions. The motion of each agent is governed by general nonlinear dynamics $\dot{x}_i = F_i(x_i, u_i)$, where $x_i \in \mathbb{R}^n$ and $u_i \in \mathbb{R}^m$ are the state, control input for the $i$-th agent, respectively and $F_i : \mathbb{R}^n \to \mathbb{R}^m$ is assumed to be locally Lipschitz continuous. Here, the vector $x_i$ consists of the position $p_i$ along with other state variables such as speed, orientation, etc. Note that it is possible to consider heterogeneous MAS where the dynamics of agents are different. However, for simplicity, we restrict our paper to the case when all the agents have the same underlying dynamics, i.e., $F_i = F$ for all $i$. The environment also consists of stationary or dynamic obstacles $\mathcal{O}_k$ for $k \in \{1, 2, \ldots, M\}$, where $\mathcal{O}_k$ represents the space occupied by obstacle $k$. The control objective for each agent is to navigate the obstacle environment to reach its goal location while maintaining safety. We use a LiDAR-based observation model similar to [33], which can be directly used for real-world robotic applications. The observation data consists of $n_{\mathrm{rays}}$ evenly-spaced rays originating at the robot and measures the *relative* locations of objects in its sensing radius. The observation data for agent $i$ is denoted by $y_i \in \mathbb{R}^{n_{\mathrm{rays}} \times \mathfrak{n}}$ where $\mathfrak{n}$ is the dimension of the environment. The collision avoidance requirement imposes that each pair of agents maintain a safety distance $2r$ where $r > 0$ is the radius of a circle that can contain the entire physical body of each agent. It also requires that each agent maintains a safe distance from other obstacles in the environment. Furthermore, each agent has a limited sensing radius $R$. We define the neighbor agents of agent $i$ as $\mathcal{N}_i^a = \{j \in V_a \mid \|p_i - p_j\| \leq R, j \neq i\}$, and the neighbor obstacles of agent $i$ as $\mathcal{N}_i^o = \{k \mid \|y_i^k\| \leq R\}$. Therefore, the agents can only sense other agents or obstacles in the set of their neighbors $\mathcal{N}_i = \mathcal{N}_i^a \cup \mathcal{N}_i^o$. The formal statement of the problem is given below.

**Problem 1** *Given a set of $N$ agents of safety radius $r$, sensing radius $R$ and a set of non-colliding goal locations $\{p_i^{\mathrm{goal}} \in \mathbb{R}^{\mathfrak{n}}\}_{i=1}^N$, design a distributed control policy $\pi_i = \pi_i(x_i, \bar{x}_i, \bar{y}_i, x_i^{\mathrm{goal}})$ for each agent $i$, where $\bar{x}_i$ is the conglomerated states of the neighbors $j \in \mathcal{N}_i^a$ and $\bar{y}_i$ the conglomerated observations from $\mathcal{N}_i^o$, such that the following holds for the closed-loop trajectories of the agents:*

- ***Collision avoidance***: *$\|y_i^j(t)\| > r, \forall j$, i.e., the agents do not collide with the obstacles; $\|p_i(t) - p_j(t)\| > 2r$ for all $t \geq 0$, $j \neq i$, i.e., the agents do not collide with each other;[2]*

- ***Liveness***: *$\inf_t \|p_i(t) - p_i^{\mathrm{goal}}\| = 0$, i.e., each agent eventually reaches its goal location $p_i^{\mathrm{goal}}$.*

## 3 Methodology

Noticing that the agents, the hitting points of LiDAR rays, and the information flow between them can be naturally modeled as a graph, we propose a novel *graph* CBF (GCBF) which encodes the collision-avoidance constraint based on the graph structure of MAS. We use a nominal controller

---

[2]In the rest of the paper, we omit the argument $t$ for the sake of brevity.

for the liveness requirement and use GNNs to learn the GCBF jointly with the collision-avoidance controller. During application, the GCBF is used to detect unsafe scenarios and switch between the nominal controller and the learned controller for collision avoidance. Our GNN architecture is capable of handling a variable number of neighbors and so, it leads to a distributed and scalable solution to the safe MAS control problem.

We start by briefly reviewing the notion of CBF commonly used in literature for safety requirements [15]. For a given closed safe set $\mathcal{S} \subset \mathbb{R}^n$, an unsafe set $\mathcal{S}_u \subset \mathbb{R}^n$ such that $\mathcal{S} \cap \mathcal{S}_u = \emptyset$, a function $h : \mathbb{R}^n \to \mathbb{R}$ is termed as a CBF if there exists a class-$\mathcal{K}$ function[3] $\alpha$ such that the following holds:

$$h(x) > 0 \ \forall x \in \text{int}(\mathcal{S}), \ h(x) < 0 \ \forall x \in \mathcal{S}_u, \ \text{and} \ \sup_u L_F h(x, u) \geq -\alpha(h(x)) \ \forall x \in \mathcal{S}, \quad (1)$$

where $L_f h(x) \coloneqq \frac{\partial h}{\partial x} f(x)$ is the Lie derivative of the function $h$ along $f$, and $\text{int}(S)$ denotes the relative interior of a closed set $S$. The existence of a CBF implies the existence of a control input $u$ which keeps the system safe. Based on the notion of CBF, we define a new notion of *graph CBF* (GCBF) for encoding safety in MAS. Before formally introducing GCBF, we briefly review the basics of the graph structure. A directed graph is defined as $\mathcal{G} = (V, E)$ where $V$ is the set of nodes and $E = \{(i_1, i_2)\}$ is the set of edges representing the flow of information from node $i_2$ to $i_1$. For the considered MAS, the nodes consist of agents $V_a$ and the hitting points $V_o$ of LiDAR rays in their observations, and hence $V = V_a \cup V_o$. The edges are defined between each of the observed points and the observing agent when the distance between them is within the sensing radius $R$. Since the flow of information is from the observed point to the observing agent, the set of edges $E = V_a \times V$. We use GNN to represent GCBF, so we first define *node* and *edge* features for GCBF.

**Node features and edge features** The nodes features $v_i$ in GCBF encode the type of the agent with $v_i = 0$ for *controlled* agents (i.e., the agents that operate under the commanded controller) and $v_i = 1$ for *uncontrolled* agents (i.e., the hitting points for LiDAR rays). The edge features $e_{ij}$ are defined as the information shared from node $j$ to agent $i$, which depends on the states of node $j$ and node $i$. Since the safety objective depends on the relative positions, one of the edge features is the relative position $p_{ij}$. The rest of the features can be chosen depending on the underlying system dynamics, e.g., relative velocities for double integrators, and relative headings for Dubin's cars. For brevity, we use $\bar{e}_i = (e_{ij_1}, e_{ij_2}, \ldots, e_{ij_{|\mathcal{N}_i|}}, \tilde{e}_{ij_{|\mathcal{N}_i|+1}}, \ldots, \tilde{e}_{ij_{N+n_{\text{rays}}}})$ with $\bar{e}_i \in \mathbb{R}^p$ for some $p > 0$, and $\bar{v}_i = (v_{j_1}, v_{j_2}, \ldots, v_{j_{|\mathcal{N}_i|}}, \tilde{v}_{j_{|\mathcal{N}_i|+1}}, \ldots, \tilde{v}_{j_{N+n_{\text{rays}}}})$ to represent the collected edge and node features for agent $i$. Here, we use $\tilde{e}_{ij}$ for $j \notin \mathcal{N}_i$ and $\tilde{v}_k$ for the rays $k \notin \mathcal{N}_i$ with constant values so that the sizes of the vectors $\bar{e}_{ij}, \bar{v}_i$ remain fixed. Now we are ready to introduce the notion of GCBF.

**Definition 1 (GCBF)** *A function $h : \mathbb{R}^p \times \{0, 1\}^{N+n_{\text{rays}}} \to \mathbb{R}$ is termed as a Graph CBF (GCBF) if there exists a class-$\mathcal{K}$ function $\alpha$ such that*

$$h(\bar{e}_i, \bar{v}_i) > 0 \ \forall x_i \in \mathcal{S}_i, \quad h(\bar{e}_i, \bar{v}_i) < 0 \ \forall x_i \in \mathcal{S}_{u,i}, \ \text{and} \ \dot{h}(\bar{e}_i, \bar{v}_i) \geq -\alpha(h(\bar{e}_i, \bar{v}_i)) \ \forall \ x_i \in \mathcal{S}_i, \quad (2)$$

*where $\mathcal{S}_i = \left\{ x_i \ \Big| \ (\|y_i^k\| > r, \forall k \in n_{\text{rays}}) \bigwedge (\min_{j \in V_a, k \neq i} \|p_i - p_j\| > 2r) \right\}$ is the safe set for agent $i$ and $\mathcal{S}_{u,i}$ the unsafe set such that $\mathcal{S}_i \cap \mathcal{S}_{u,i} = \emptyset$.*

Note that we use GNN to model GCBF with input $(e_{ij_1}, \ldots, e_{ij_{|\mathcal{N}_i|}})$ and $(v_{j_1}, \ldots, v_{j_{|\mathcal{N}_i|}})$ since GNN can have variable-size inputs. Since it is not possible to mathematically define a function with a variable-size input, we use fixed-size input $(\bar{e}_i, \bar{v}_i)$ so that we can define the input domain of function $h$. Since the node features are constant, $\dot{h}$ is computed with respect to the edge features as

$$\dot{h}(\bar{e}_i, \bar{v}_i) = \frac{\partial h(\bar{e}_i, \bar{v}_i)}{\partial \bar{e}_i} \frac{\partial \bar{e}_i}{\partial x_i} F(x_i, u_i) + \frac{\partial h(\bar{e}_i, \bar{v}_i)}{\partial \bar{e}_i} \sum_{j \in \mathcal{N}_i} \frac{\partial \bar{e}_i}{\partial x_j} F(x_j, u_j). \quad (3)$$

Note that while choosing the edge features $\bar{e}_i$, it is important to make sure that the time derivates of the features of agent $i$ include the control $u_i$ so that the input can help keep the system safe. Thus, we assume $\frac{\partial}{\partial u_i} \left( \frac{\partial h(\bar{e}_i, \bar{v}_i)}{\partial \bar{e}_i} \frac{\partial \bar{e}_i}{\partial x_i} F(x_i, u_i) \right) \not\equiv 0$. This is similar to assuming $L_g h \not\equiv 0$ for CBF in (1) where

---

[3] A monotonically increasing continuous function $\alpha : \mathbb{R}_+ \to \mathbb{R}_+$ with $\alpha(0) = 0$ is termed as class-$\mathcal{K}$.

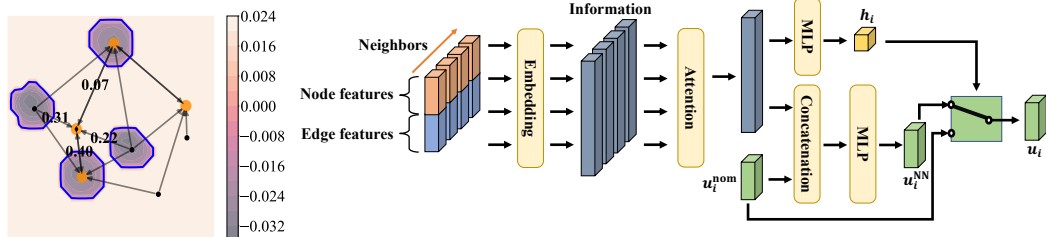

Figure 1: Left: the contours of the learned GCBF of the agent with a diamond mark, where the blue boundary is the 0-level set of the GCBF. Orange circles are agents and black dots are obstacles. The weights on the edges show the attention values. Right: the overview of the proposed framework.

$F(x, u) = f(x) + g(x)u$, which is very common (see [15]). Under this assumption, we can state the following result on the safety of the system under GCBF (the proof is given in Appendix C.1).

**Theorem 1** *Given a set of $N$ agents, assume that there exists a GCBF $h$ satisfying (2) for some class-$\mathcal{K}$ function $\alpha$. Then, the resulting closed-loop trajectories of agents with non-colliding initial conditions under any smooth control input $u_i \in \mathcal{U}_i^{safe} := \left\{ u \in \mathcal{U}_i \mid \dot{h}(\bar{e}_i, \bar{v}_i) + \alpha(h(\bar{e}_i, \bar{v}_i)) \geq 0 \right\}$ satisfy $x_i(t) \in \mathcal{S}_i$ for all $i \in V_a$ and $t \geq 0$.*

**Safe control policy** For the multi-objective Problem 1, we use a hierarchical approach for the goal-reaching and the safety objectives. First, we design a nominal controller $u_i^{\text{nom}} = \pi_{\text{nom}}(x_i, x_i^{\text{goal}})$ for the goal-reaching objective. In this work, we use LQR and PID-based nominal controllers. Next, using the nominal controller, we design a minimum-norm controller that satisfies the safety constraint using an optimization framework. With GCBF $h$, a solution to the following optimization problem:

$$\min_{u_i \in \mathcal{U}_i} \quad \|u_i - u_i^{\text{nom}}\|^2, \tag{4a}$$

$$\text{s.t.} \quad \frac{\partial h(\bar{e}_i, \bar{v}_i)}{\partial \bar{e}_i} \frac{\partial \bar{e}_i}{\partial x_i} F(x_i, u_i) + \sum_{j \in \mathcal{N}_i} \frac{\partial h(\bar{e}_i, \bar{v}_i)}{\partial \bar{e}_i} \frac{\partial \bar{e}_i}{\partial x_j} F(x_j, u_j) \geq -\alpha(h(\bar{e}_i, \bar{v}_i)), \tag{4b}$$

keeps agent $i$ in its safety region. Note that (4) is *not a distributed framework* for finding the control policy, since the constraint for computing $u_i$ depends on $u_j$. Thus, it is not straightforward to solve (4) in a distributed manner, although there is some work on addressing such problems [54]. To this end, we use an NN-based control policy that satisfies the safety constraint and does not require solving a centralized non-convex optimization problem online. Next, we discuss the training setup for jointly learning both GCBF and a distributed safe control policy (see Figure 1).

**Neural GCBF and distributed control policy training** We parameterize a candidate GCBF $h_\theta$ as NNs with parameters $\theta$. The NN contains a GNN backbone and a multilayer perceptron (MLP) head. In the backbone, each connected edge $\{i, j\}$ first goes through an MLP $f_{\theta_1}$, which encodes the edge feature $e_{ij}$ and the node features $v_j$ to latent space, i.e., $q_{ij} = f_{\theta_1}(e_{ij}, v_j)$. Then, we use attention [55] to aggregate the information of the neighbors, i.e., $q_i = \sum_{j \in \mathcal{N}_i} \text{softmax}(f_{\theta_2}(q_{ij}))f_{\theta_3}(q_{ij})$, where $f_{\theta_2}$ and $f_{\theta_3}$ are two NNs parameterized by $\theta_2$ and $\theta_3$. $f_{\theta_2}$ is often called "gate" NN in literature [56], and the output of $\text{softmax}(f_{\theta_2}(q_{ij}))$ is called "attention", which is a scatter value between $0$ and $1$ for each agent $j \in \mathcal{N}_i$ represents how critical agent $j$ is to agent $i$. We discuss later the necessity of applying attention. After the backbone, aggregated information is processed by the head with parameters $\theta_4$ to get the GCBF value for each agent, i.e., $h_i = f_{\theta_4}(q_i)$.

We design the neural distributed controller as $u_i^{\text{NN}} = \pi_\phi(\bar{e}_i, \bar{v}_i, \pi_{\text{nom}}(x_i, x_i^{\text{goal}}))$. The distributed control policy $\pi_\phi$ is an NN with a similar structure as GCBF, designed for collision and obstacle avoidance. The GNN backbone of $\pi_\phi$ is the same as the GNN backbone of $h_\theta$, except that $\pi_\phi$ also uses $\pi_{\text{nom}}$ as its feature. This helps the NN controller learn how to modify the agent's behavior given a nominal policy to avoid collisions with the neighbors and obstacles. Thus, we concatenate the nominal control signal $u_i^{\text{nom}}$ with the output of the GNN component as the input to the MLP component of $\pi_\phi$ (see Figure 1). Note that the input to the control policy is only the local information

$(\bar{e}_i, \bar{v}_i)$, and unlike (4), it does not require knowledge of neighbors' inputs. In this way, the controller is fully distributed, and thanks to GNN's ability to handle variable sizes of inputs, $\pi_\phi$ generalizes to larger graphs with much more neighbors. We train the GCBF and the distributed controller by minimizing the empirical loss $\mathcal{L} = \sum_{i \in V_a} \mathcal{L}_i$, where $\mathcal{L}_i$ is the loss for agent $i$ defined as

$$
\begin{aligned}
\mathcal{L}_i(\theta, \phi) = &\sum_{x_i \in \mathcal{S}_i} \left[ \gamma - \dot{h}_\theta(\bar{e}_i, \bar{v}_i) - \alpha(h_\theta(\bar{e}_i, \bar{v}_i)) \right]^+ + \sum_{x_i \in \mathcal{S}_i} [\gamma - h_\theta(\bar{e}_i, \bar{v}_i)]^+ \\
&+ \sum_{x_i \in \mathcal{S}_{u,i}} [\gamma + h_\theta(\bar{e}_i, \bar{v}_i)]^+ + \eta \left\| \pi_\phi(\bar{e}_i, \bar{v}_i, \pi_{\text{nom}}(x_i, x_i^{\text{goal}})) - \pi_{\text{nom}}(x_i, x_i^{\text{goal}}) \right\|,
\end{aligned}
\tag{5}
$$

where $[\cdot]^+ = \max(\cdot, 0)$, and $\gamma > 0$ is a hyper-parameter to encourage strict inequalities. The training data $\{x_i\}$ is collected over multiple scenarios and the loss is calculated by evaluating the CBF conditions on each sample point. We use the on-policy strategy to collect data by executing the learned controller $\pi_\phi$ with probability $1 - \epsilon$, and the nominal controller $\pi_{\text{nom}}$ with probability $\epsilon$, where $\epsilon$ decreases linearly from 1 to 0 during the training. In this way, we avoid the low-quality data generated by $u_i^{\text{NN}}$ at the beginning of the training and match the train-test distributions at the end of the training. The first three terms in (5) correspond to the GCBF conditions in (2), while the last term encourages small control deviation from $\pi_{\text{nom}}$ so that $\pi_\phi$ can have better goal-reaching performance with $\eta > 0$ to balance the weight of the GCBF constraint losses and the norm of the resulting input.

One of the challenges of evaluating the loss function $\mathcal{L}$ is estimating $\dot{h}_\theta$. Similar to [21], we estimate $\dot{h}_\theta$ by $(h_\theta(\bar{e}_i(t_{k+1}), \bar{v}_i(t_{k+1})) - h_\theta(\bar{e}_i(t_k), \bar{v}_i(t_k)))/\delta t$, where $\delta t = t_{k+1} - t_k$ is the simulation timestep. However, the discretized approximation may cause issues if the graph connections change between any two consecutive time steps. Fortunately, the attention mechanism we use naturally addresses this problem. During training, the agents learn to pay more attention (i.e., close to 1) to nodes that are near and less attention (i.e., close to 0) to the nodes that are at the boundary of the sensing region. Therefore, if an edge breaks in between time steps and a node gets out of the sensing radius, the CBF value does not change significantly. In this manner, the estimation of $\dot{h}_\theta$ does not encounter large errors due to edge changes between time steps. Note that $\dot{h}_\theta$ includes the inputs from agent $i$ as well as the neighbor agents $j \in \mathcal{N}_i$. Therefore, during training, when we use gradient descent and backpropagate $\mathcal{L}_i(\theta, \phi)$, the gradients are passed to not only the controller of agent $i$ but also the controllers of all neighbors in $\mathcal{N}_i$.[4] For the class-$\mathcal{K}$ function $\alpha$, we simply use $\alpha(h) = \alpha \cdot h$, where $\alpha > 0$ is a constant. More details on the training process are provided in Appendix A.1, and a discussion on the generalization result of neural GCBF is provided in Appendix C.2.

**GCBF detector and online policy refinement** When the training finishes, we can execute our controller in a fully distributed manner. To achieve better goal-reaching performance, we use the learned GCBF as a detector to detect unsafe scenarios and use a switching control policy to reduce potential conservatism due to only using learned policy $\pi_\phi$. In particular, we define the control assignment for each agent as $u_i = u_i^{\text{nom}}$ if $u_i^{\text{nom}} \in \mathcal{U}_i^{\text{safe}}$ and $u_i = u_i^{\text{NN}}$, otherwise. Namely, at each time step, the system uses the nominal controller if the GCBF conditions (2) are satisfied with the nominal controller $u_i^{\text{nom}}$. If not, it switches to the learned policy $u_i^{\text{NN}}$ for collision avoidance.

While the control policy $\pi_\phi$ is designed to satisfy the GCBF conditions (2), the GCBF conditions can still be violated because of various reasons, such as distribution shift in testing and difficulty in exploring the state-space in high-dimensional and large-scale MAS during training. To this end, similar to [20], we use an online policy refinement technique to make the learned policy *safer*. At a given time instant, if the learned policy $\pi_\phi$ does not satisfy the GCBF conditions (2), we compute the residue $\delta(u_i^{\text{NN}}) = \max\left(0, \gamma - \dot{h}_\theta(\bar{e}_i, \bar{v}_i) - \alpha(h_\theta(\bar{e}_i, \bar{v}_i))\right)$ and use gradient descent to update the control policy $\pi_\phi$ until $\delta(u_i^{\text{NN}}) = 0$ or the maximum iteration is reached.

## 4 Experiments

**Environments** We conduct experiments on three different environments consisting of a SimpleCar modeled under double-integrator dynamics, a DubinsCar model, and a Drone modeled under linearized drone dynamics (see Appendix A.2 for more details). Both car environments are 2D while the drone environment is 3D. The parameters are $R = 1, r = 0.05, u_M = 0.8$ in the 2D environments,

---

[4]We re-emphasize on the fact that during testing, the neighbors' inputs are not required for $\pi_\phi$.

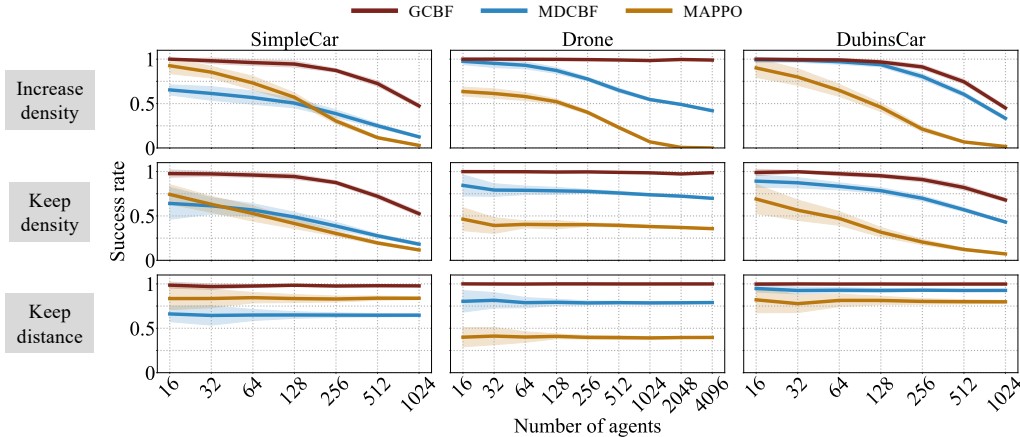

Figure 2: Success rate of GCBF, MDCBF, and MAPPO algorithms across the three environments and the three sets of experiments, namely, increasing density of the agents in a fixed workspace, increasing the size of the workspace to keep the density same, and increasing the size of the workspace but limiting the average distance traveled by agents.

and $R = 0.5, r = 0.05, u_M = 0.6$ for the 3D environment, where $u_M$ is the maximum speed of the agents. The workspace $\mathcal{X} = l^n$ of each of the environments is a hyper-rectangle of side-length $l > 0$. The total timesteps of experiments are 2500 for 2D environments and 2000 for 3D environments.

**Evaluation criteria** We use safety rate, reaching rate, and success rate as the evaluation criteria for the performance of a chosen algorithm. The safety rate is defined as the ratio of agents not colliding with either obstacles or other agents during the experiment period over all agents. The reaching rate is defined as the ratio of agents reaching their goal location by the end of the experiment period. The success rate is defined as the ratio of agents that are both safe and goal-reaching. We note that the safety metric in [21] is slightly misleading as they measure the portion of collision-free states for safety rate. For each environment, we evaluate the performance over 16 instances of randomly chosen initial and goal locations from the workspace $\mathcal{X}$ for 3 policies trained with different random seeds. Here, we report the mean success rate and their standard deviations for the 16 instances for each of the 3 policies. We report the safety rate, reaching rate, and ablation results in Appendix B.

**Baselines** We use MDCBF [20] and MAPPO [14] as the baselines for comparisons (see Appendix A.3 for implementation details of the baselines). MDCBF learns pair-wise CBFs between agents and takes the minimum on one agent as the CBF value of this agent. Furthermore, it considers each neighbor equally important without attention and does not use CBF as a detector but directly uses the learned controller. MAPPO is a MARL-based algorithm that learns to be safe and goal-reaching by maximizing the expected reward. For fair comparisons, we re-implement MAPPO using GNN.

**Experiment settings** We conduct four sets of experiments to demonstrate the scalability, generalizability, and reliability of the proposed method. First, we fix the workspace size $\mathcal{X}$ where the agent trajectories evolve. In this experiment, we use $\mathcal{X} = 32 \times 32$ for 2D car environments and $\mathcal{X} = 16 \times 16 \times 16$ for the 3D drone environment and perform experiments with up to 1024 agents for the 2D environments and up to 4096 agents for the 3D environment. In the second set of experiments, we keep the per-unit agent density constant. To this end, we increase the size of $\mathcal{X}$ as the number of agents increases from 16 to 4096 (see Appendix A.2 for workspace sizes). In the third set of experiments, we further constrain the maximum traveling distance to 4.0 units for each agent while increasing the size of the workspace to keep the per-unit agent density constant. In the fourth set of experiments, we introduce moving obstacles where we perform experiments in the DubinsCar environment with up to 32 obstacles and 64 agents in a workspace $\mathcal{X} = 12 \times 12$. The obstacles are assumed to be moving with a bounded, constant, unknown speed up to 0.2 units and the size of the obstacle varies between 0 to 0.5 units. Agents use LiDAR to detect obstacles. Each agent generates equally-spaced 32 rays with a maximum sensing radius $R = 1.0$ unit. For the first three experiments, we train all the algorithms with 16 agents, and for the fourth experiment, we train with 64 agents and with 16 randomly generated point-sized obstacles to model LiDAR observations.

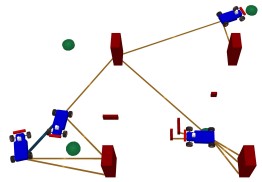 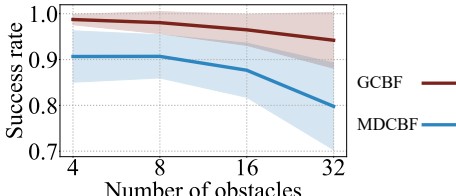

Figure 3: Left: Illustration of the DubinsCar environment with obstacles. The green circles are goal points and the red rectangles are obstacles. The solid blue line shows the connection between agents and the orange lines show LiDAR rays. Right: Success rate plots for GCBF and MDCBF.

**Results** Figure 2 shows the performance of the proposed framework (GCBF) against the baselines MDCBF and MAPPO. In all the experiments, the success rate of GCBF is higher than that of the considered baselines. Particularly as the number of agents increases, the decrement in the success rate of MAPPO and MDCBF is very high. For the SimpleDrone environment, we notice that there is almost no drop in the success rate with an increase in the number of agents. We speculate that this is because the agents in 3D have more degrees of freedom to move to avoid collisions and hence, achieve a very high safety rate (see the individual safety and goal-reaching plots in Appendix B). For the first two sets of experiments, the success rate drop is primarily because the inter-agent interactions are increasing. In the first set of experiments, it is clear with an increase in the density of agents for a fixed workspace, the inter-agent interactions increase. For the second set of experiments, although the per-unit agent density is the same, with an increase in the workspace size, the average distances traveled by the agents in randomly generated initial and goal location instances also increase. Thus, the inter-agent interaction increases. We designed the third set of experiments to further analyze the effect of traveling distance on success rate. In the third set of experiments, not only the density but also the average distance traveled by each agent is fixed, which keeps the number of inter-agent interactions constant. The success rate of GCBF remains very close to 1 with minimum success rate of GCBF is 97.1% for SimpleCar, 99.7% for DubinsCar, and 99.5% for SimpleDrone. Figure 3 illustrates that the proposed method using GCBF achieves a higher success rate across obstacle environments as compared to MDCBF since it can deal with different types of neighbors. The success rate of MAPPO with obstacles is consistently lower than 0.1, so we do not include it in the plot.

## 5 Limitations

In the current framework, there is no cooperation among the controlled agents, which leads to conservative behaviors. In certain scenarios, this can also lead to deadlocks. Another limitation is the assumption of knowledge of the neighbors' velocities. From a practical point of view, measuring relative position is possible using LiDAR or other sensors, but accurate estimation of other agents' velocities and accelerations is not possible. Similar to any other NN-based control policy, the proposed method also suffers from difficulty in providing formal guarantees of correctness. In particular, it is difficult, if not impossible, to verify that the proposed algorithm can always keep the system safe via formal verification of the learned neural networks (see Appendix 1 in [57] on NP-completeness of NN-verification problem). These limitations inform our future line of work on relaxation of the assumption on available information, introducing cooperation among agents to reduce conservatism, and looking into methods of verification of the correctness of the control policy.

## 6 Conclusions

In this paper, we introduce a new notion of GCBF to encode inter-agent collision and obstacle avoidance in control for large-scale multi-agent systems with LiDAR-based observations, and jointly learn it with a distributed controller using GNNs. The proposed control framework is completely distributed as each agent only uses local information in its sensing region, and thus, is scalable to large-scale problems. Experimental results demonstrate that even when trained on small-scale MAS, the proposed method can achieve higher success rates in completing goal-reaching tasks while maintaining safety for large-scale MAS even in the presence of dynamic obstacles.

**Acknowledgments**

This work was partly supported by the National Science Foundation (NSF) CAREER Award #CCF-2238030 and the MIT-DSTA program. Any opinions, findings, conclusions or recommendations expressed in this publication are those of the authors and don't necessarily reflect the views of the sponsors.

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

# Appendix

## A Experiment details

In this section, we introduce the details of the experiments, including the implementation details of GCBF and the baselines, the dynamics of the agents, and the settings of the environments.

### A.1 Implementation details

Our learning framework contains two neural network models: the GCBF $h_\theta$ and the controller $\pi_\phi$, both of which contain a GNN component and an MLP component. The GNN component of $h_\theta$ consists of three NNs: the embedding NN $f_{\theta_1}$, the NN used in the attention aggregation $f_{\theta_2}$ and $f_{\theta_3}$. $f_{\theta_1}$ is a two-hidden-layer MLP with 2048 neurons each, which maps the input to a 256D information space. The gate NN $f_{\theta_2}$ is a two-hidden-layer MLP with 128 neurons each. $f_{\theta_3}$ is a two-hidden-layer MLP with 2048 neurons each, which maps the information to a 1024D space. The output of the GNN component of $h_\theta$ goes through another MLP component $f_{\theta_4}$, which is a three-hidden-layer MLP with neurons 512, 128, 32 that generates the GCBF value. In summary,

$$h_i = h_\theta \left([v_j; e_{ij}]|_{j \in \mathcal{N}_i}\right) = f_{\theta_4} \circ \left( \sum_{j \in \mathcal{N}_i} \text{softmax}(f_{\theta_2} \circ (f_{\theta_1}([v_j; e_{ij}]))) \cdot f_{\theta_3} \circ (f_{\theta_1}([v_j; e_{ij}]))\right). \quad (6)$$

To make the training easier, we define $\pi_\phi = \pi_\phi^{\text{NN}} + \pi_{\text{nom}}$, where $\pi_\phi^{\text{NN}}$ is the NN controller and $\pi_{\text{nom}}$ is the nominal controller. In this way, $\pi_\phi^{\text{NN}}$ only needs to learn the deviation from $\pi_{\text{nom}}$. The controller $\pi_\phi^{\text{NN}}$ uses the same GNN component as $h_\theta$, but before passing the output of the GNN component to the MLP component, we concatenate the output of the GNN component with the output of the nominal controller $\pi_{\text{nom}}$ and pass the concatenated vectors to the MLP component to get the final output of the controller (see Figure 1).

We use Adam [58] with a learning rate $3 \times 10^{-4}$ for $h_\theta$ and $1 \times 10^{-3}$ for $\pi_\phi$ to optimize the NNs for $500,000$ steps in training. The training time is around 60 minutes on a 13th Gen Intel(R) Core(TM) i7-13700KF CPU @ 3400MHz and an NVIDIA RTX 3090 GPU. In training, we time each loss term with a balancing coefficient, i.e.,

$$\mathcal{L}_i(\theta, \phi) = \eta_{\text{safe}} \sum_{x_i \in \mathcal{S}_i} \max\left(0, \gamma - h_\theta(\bar{e}_i, \bar{v}_i)\right) + \eta_{\text{unsafe}} \sum_{x_i \in \mathcal{S}_{u,i}} \max\left(0, \gamma + h_\theta(\bar{e}_i, \bar{v}_i)\right)$$
$$+ \eta_{\text{deriv}} \sum_{x_i \in \mathcal{S}_i} \max\left(0, \gamma - \dot{h}_\theta(\bar{e}_i, \bar{v}_i) - \alpha(h_\theta(\bar{e}_i, \bar{v}_i))\right) + \eta \|\pi_\phi(\bar{e}_i, \bar{v}_i, \pi_{\text{nom}}(x_i, x_i^{\text{goal}}))\|,$$
$$(7)$$

and we choose the hyper-parameters following Table 1.

Table 1: Hyper-parameters used in our training

| Environment | $\alpha$ | $\gamma$ | $\eta_{\text{safe}}$ | $\eta_{\text{unsafe}}$ | $\eta_{\text{deriv}}$ | $\eta$ |
|---|---|---|---|---|---|---|
| SimpleCar | 1.0 | 0.02 | 1.0 | 1.0 | 0.5 | 0.05 |
| Drone | 1.0 | 0.02 | 1.0 | 1.0 | 0.5 | 0.05 |
| DubinsCar | 1.0 | 0.02 | 1.0 | 1.0 | 0.2 | 0.0001 |

In practice, we find that it benefits training if there is a non-empty region between the safe and the unsafe regions as it provides the NNs with some flexibility to fit the safe-unsafe boundary. Thus, we define the region to be unsafe if the distance between two agents is less than $2r$, or the agents hit the obstacle, and safe if the distance between two agents or an agent and an obstacle is more than $4r$.

### A.2 Environments

Table 2 provides the side length $l$ for the experiments with keeping density. Next, we provide the details of each experiment environment.

**SimpleCar** We use double integrator dynamics for the SimpleCar environment. The state of agent $i$ is given by $x_i = [p_i^x, p_i^y, v_i^x, v_i^y]^\top$, where $[p_i^x, p_i^y]^\top$ is the position of the agent, and $[v_i^x, v_i^y]^\top$ is

Table 2: Side-length $l$ of the workspace $\mathcal{X} = l^n$ for various $N$.

| $N$ | 16 | 32 | 64 | 128 | 256 | 512 | 1024 | 2048 | 4096 |
|---|---|---|---|---|---|---|---|---|---|
| 2D | 8 | 11.3 | 16 | 22.6 | 32 | 45.3 | 64 | N/A | N/A |
| 3D | 6.35 | 8 | 10.1 | 12.7 | 16 | 20.2 | 25.4 | 32 | 40.3 |

the velocity. The action of agent $i$ is given by $u_i = [a_i^x, a_i^y]^\top$, i.e., the acceleration. The dynamics function is given by:

$$\dot{x}_i = \begin{bmatrix} v_i^x \\ v_i^y \\ a_i^x \\ a_i^y \end{bmatrix} \tag{8}$$

The simulation timestep is $\delta t = 0.03$. In this environment, we use $e_{ij} = x_j - x_i$ as the edge information. For training MAPPO, we design the reward in this way: in each timestep, the agents receive a $-0.01 - 0.0001\|u_i\|$ reward. The agents also receive a $-2$ reward for collision at each timestep and a $+4$ reward for reaching the goal.

**DubinsCar** We use the standard Dubin's car model in this environment. The state of agent $i$ is given by $x_i = [p_i^x, p_i^y, \theta_i, v_i]^\top$, where $[p_i^x, p_i^y]^\top$ is the position of the agent, $\theta_i$ is the heading, and $v_i$ is the speed. The action of agent $i$ is given by $u_i = [\omega_i, a_i]^\top$ containing angular velocity and longitudinal acceleration. The dynamics function is given by:

$$\dot{x}_i = \begin{bmatrix} v_i \cos(\theta_i) \\ v_i \sin(\theta_i) \\ \omega_i \\ a_i \end{bmatrix} \tag{9}$$

The simulation timestep is $\delta t = 0.03$. We use $e_{ij} = e_j(x_j) - e_i(x_i)$ as the edge information, where $e_i(x_i) = [p_i^x, p_i^y, v_i \cos(\theta_i), v_i \sin(\theta_i), \theta_i]^\top$. For training MAPPO, we design the reward in this way: in each timestep, the agents receive a $-0.0001 - 0.01\|u_i\|$ reward. The agents also receive a $-0.1$ reward for collision at each timestep and a $+10$ reward for reaching the goal.

**SimpleDrone** We use a linearized model for drones in our experiments. The state of agent $i$ is given by $x_i = [p_i^x, p_i^y, p_i^z, v_i^x, v_i^y, v_i^z]^\top$ where $[p_i^x, p_i^y, p_i^z]^\top$ is the 3D position, and $[v_i^x, v_i^y, v_i^z]^\top$ is the 3D velocity. The control inputs are $u_i = [a_i^x, a_i^y, a_i^z]^\top$, and the dynamics function is given by:

$$\dot{x}_i = \begin{bmatrix} v_i^x \\ v_i^y \\ v_i^z \\ -1.1v_i^x + 1.1a_i^x \\ -1.1v_i^y + 1.1a_i^y \\ -6v_i^z + 6a_i^z \end{bmatrix} \tag{10}$$

The simulation timestep is $\delta t = 0.03$. We use $e_{ij} = x_j - x_i$ as the edge information. For training MAPPO, we design the reward in this way: in each timestep, the agents receive a $-0.01 - 0.001\|u_i\|$ reward. The agents also receive a $-1$ reward for collision at each timestep and a $+10$ reward for reaching the goal.

In each case, the simulation terminates when all the agents reach the goal.

### A.3 Baseline methods

We implement MDCBF [20] based on the official implementation[5], and MAPPO [14] based on the official implementation[6]. For fair comparisons, we changed the structure of the actor and the critic to GNNs. The training time of GCBF and MDCBF are similar, both about 60 minutes, and around 5 hours for MAPPO, on a 13th Gen Intel(R) Core(TM) i7-13700KF CPU @ 3400MHz and an NVIDIA RTX 3090 GPU.

---

[5]https://github.com/MIT-REALM/macbf
[6]https://github.com/zoeyuchao/mappo

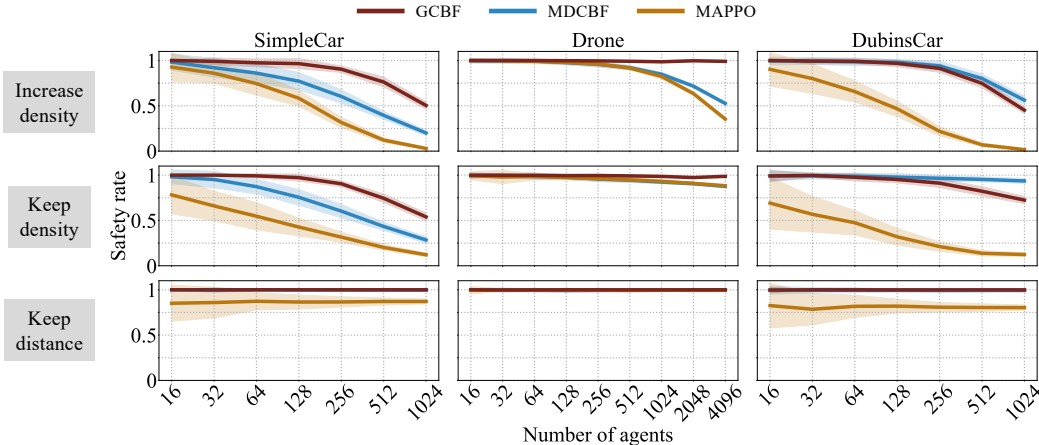

Figure 4: Safety rate of GCBF, MDCBF, and MAPPO algorithms across the three environments and the three sets of experiments, namely, increasing density of the agents in a fixed workspace, increasing the size of the workspace to keep the density same, and increasing the size of the workspace but limiting the average distance traveled by agents. Note that there are several overlaps in this figure: MDCBF with GCBF in the left bottom figure, the middle figure, and the right bottom figure; and all three lines in the middle bottom figure.

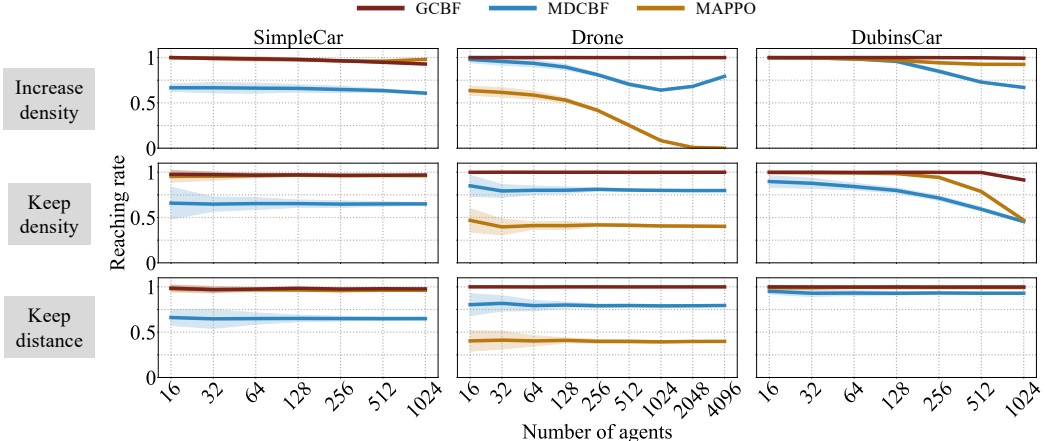

Figure 5: Goal-reaching rate of GCBF, MDCBF, and MAPPO algorithms across the three environments and the three sets of experiments, namely, increasing density of the agents in a fixed workspace, increasing the size of the workspace to keep the density same, and increasing the size of the workspace but limiting the average distance traveled by agents. Note that there are several overlaps of the lines: GCBF and MAPPO in the left three figures and the right bottom figure.

## B  Supplementary Experimental Results

### B.1  Safety and Goal-reaching rates

We include the plots of the safety rates and goal-reaching rates for the first three sets of experiments (increase density, keep density, and keep distance) in Figure 4 and Figure 5, respectively. In the SimpleCar environment, GCBF achieves a higher safety rate in all three sets of experiments, while in the Drone environment, the performance of the baselines and GCBF overlap for the third set of experiments where we keep the average traveling distance of agents constant. However, As can be observed in Figure 5, the goal-reaching rate of GCBF is higher than MDCBF and MAPPO, which results in a higher success rate as depicted in Figure 2 in the main paper. Similarly, for the DubinsCar environment, MDCBF achieves similar or better safety than GCBF, but it has a lower

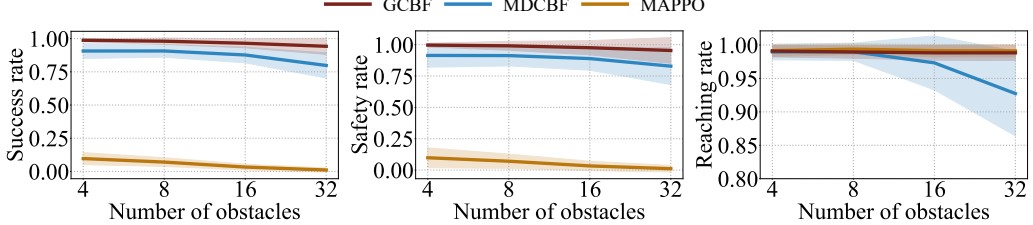

Figure 6: Left: success rate plots for GCBF and MDCBF. Middle: safety rate plots for GCBF and MDCBF. Right: goal-reaching rate plots for GCBF and MDCBF. Note that MAPPO and GCBF overlap in the reaching rate plot on the right.

goal-reaching rate, resulting in a lower overall success rate. On the other hand, MAPPO achieves very high goal-reaching rates in both SimpleCar and DubinsCar environments but fails to achieve a high safety rate. This corroborates the reasoning mentioned in the Introduction Section in the main paper that MARL-based methods use penalties for safety violations and thus, cannot achieve a high safety rate in practice. We emphasize that success rate is a better metric of performance evaluation since there is always a trade-off between prioritizing goal-reaching objectives and safety objectives, and success rate provides a fair method of evaluating both at the same time.

For the obstacle environment, we present the success, safety, and goal-reaching rates for all three algorithms, namely, GCBF, MDCBF, and MAPPO in Figure 6. In the presence of obstacles also, the goal-reaching rate is very high for MAPPO, but the safety rate is close to zero. Furthermore, we can observe a sharp drop in the performance of MDCBF with 32 obstacles, while that of GCBF does not change significantly. We remind the reader that the training is performed with 16 point-sized obstacles to model LiDAR observations, and conduct experiments with 32 large-sized experiments. Thus, a high success rate in all the test cases illustrates the generalizability of GCBF.

Additionally, we conduct tests for GCBF with an increasing number of agents as well as obstacles, while keeping the ratio of the number of agents to the number of obstacles constant. We fix this ratio as 4 and report the performance in Figure 7. It can be observed that the safety rate remains close to 1 for all cases, with the minimum safety rate being 99.12%. The success rate drops to 84% for 128 agents. We believe that this is because the workspace becomes very crowded with 128 agents and 32 obstacles, and hence, the agents are unable to reach their goals. However, the high safety rate illustrates that the proposed method can keep the large-scale system safe even with a large number of obstacles.

## B.2  Ablation studies

**Ablation on sensing region**  We perform ablation experiments to study the effect of the sensing radius parameter $R$ on the success rate. We train GCBF for the DubinsCar environment with $R = 1$ and perform tests with $R = [0.05, 0.1, 0.2, 0.5, 0.75, 1]$ (note that the safety distance is $r = 0.05$). We observe that the success rate is lower for smaller values of $R$, as expected since a smaller sensing radius implies that agents have less time to react to the neighbors. Also, it can be observed from the left plot in Figure 8 that the success rate saturates at $R = 0.5$, which implies that the trained GCBF is robust to the sensing radius and can have similar performance even with much smaller sensing region.

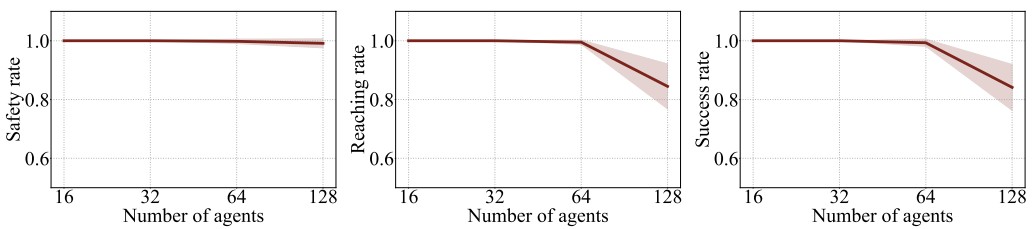

Figure 7:  GCBF experiments in obstacle environment with an increasing number of agents. Left: success rate. Middle: safety rate plots. Right: goal-reaching rate plots.

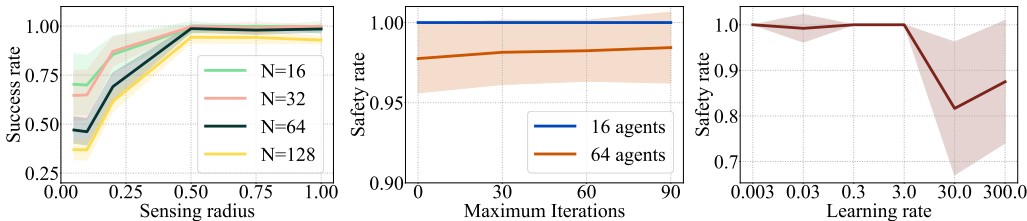

Figure 8: Ablation study for sensing radius $R$ (left), maximum iterations of online policy update (middle), and learning rate for online policy update (right).

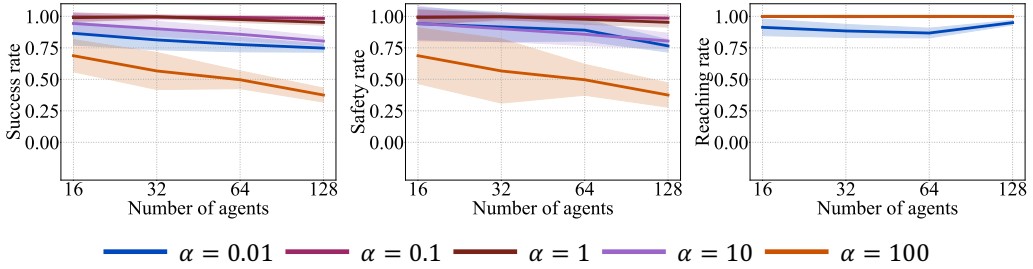

Figure 9: Ablation study for the class-$\mathcal{K}$ parameter $\alpha$.

**Ablation on online policy refinement** Similar to [20], we perform ablation studies on online policy refinement for improved safety. In the first set of experiments, we vary the maximum iteration for policy refinement from 0 to 90 for a fixed learning rate of $lr = 0.3$ and plot the safety rates in the middle plot in Figure 8. It can be observed that for 16 agents, policy refinement is not needed as even without policy refinement, the safety rate is 1.0. For 64 agents, online policy refinement improves the safety rate from 0.977% to 0.984%. This illustrates that online policy refinement can help improve the safety rate, especially as the number of agents grows.

The effect of the learning rate on the online policy update is captured in the right plot in Figure 8. Here, we observe that for learning rate $lr \leq 3$, the safety rate remains high, but it drops significantly for higher learning. Thus, in conclusion, there is a wide range of hyper-parameters for which GCBF works with a high success rate, and its performance is robust to changes in these hyper-parameters.

**Ablation of class-$\mathcal{K}$ parameter** $\alpha$ We also perform ablation studies on the class-$\mathcal{K}$ parameter $\alpha$ in the CBF derivative condition (the third inequality in (2)). We train and test GCBF for various choices of the parameter $\alpha$ from the set $[0.01, 0.1, 1, 10, 100]$. The testing performance for different values of $\alpha$ for various numbers of agents for the DubinsCar environment. It can be observed that the GCBF performance peaks for $\alpha = 1.0$ and it performs relatively well for $\alpha = 0.1$ as well. As the parameter $\alpha$ increases, the performance decays. The primary reason for that, according to our intuition, is that larger $\alpha$ in a discrete-time simulation can lead to larger control inputs that can lead to violation of safety even if the continuous-time CBF conditions are satisfied. Furthermore, a very small value of $\alpha$ (in this case, $\alpha = 0.01$) leads to a very conservative approach, and the goal-reaching rate drops.

**Website** We include other details on our project website https://mit-realm.github.io/gcbf-website/.

### B.3 Handcrafted CBF

We conducted further experiments with a handcrafted CBF for the SimpleCar experiments as it is relatively easier to design a handcrafted CBF for a double-integrator system. We design the CBF between each pair of agents as

$$h(x_1, x_2) = 2(p_1^x - p_2^x)(v_1^x - v_2^x) + 2(p_1^y - p_2^y)(v_1^y - v_2^y) + \left[(p_1^x - p_2^x)^2 + (p_1^y - p_2^y)^2 - 4r^2\right]. \quad (11)$$

We design two frameworks:

- Centralized CBF: In this framework, inputs of all the agents are solved together by setting up a centralized QP containing CBF constraints of all the agents.

Table 3: Safety rates and the computational time (second) per control step of the handcrafted CBF

| Number of agents | Safety rate | | Computational time | |
|---|---|---|---|---|
| | Centralized | Decentralized | Centralized | Decentralized |
| 16 | 1.0 | 0.945 | 0.009 | 0.00047 |
| 32 | 1.0 | 0.820 | 0.013 | 0.00064 |
| 64 | 1.0 | 0.734 | 0.039 | 0.00096 |
| 128 | 1.0 | 0.550 | 0.244 | 0.00156 |

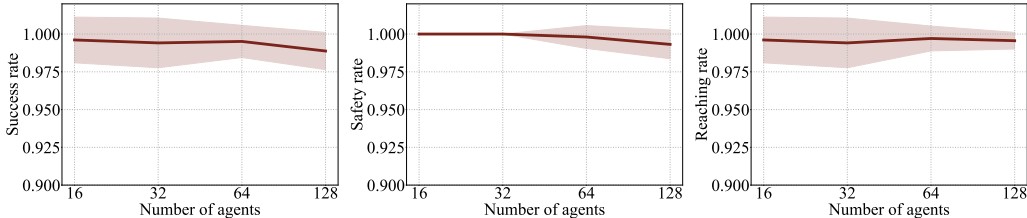

Figure 10: Success rate, safety rate, and goal reaching rate of GCBF in the CrazyFlie environment.

- Decenteratlized CBF: In this framework, each agent computes its own control input through a QP with CBF constraint for its safety and assumes the neighbors take the nominal control $\pi_{\mathrm{nom}}$ as it is not possible to obtain the current control action in a synchronous setup.

We show the safety rate results and the computational time (second) per control step in Table 3. Our experiments illustrated that the centralized hand-crafted CBF-QP-based controller can maintain a safety rate of 1.0. However, it is computationally very costly to solve CBF-QP for a large-scale MAS. In particular, we found out that for up to 64 agents, the CBF-QP can be used for "real-time" control synthesis (discrete-time step of 0.03s), but for more agents, the CBF-QP is too slow (0.244s per step for 128 agents) and cannot be used for "real-time" control synthesis. We also tried to increase the control time step to 0.25s for 128 agents so that the CBF-QP is "real-time", but the safety rate drops to 0.55. The decentralized QP, on the other hand, is fast enough to be used for real-time control synthesis for a large number of agents. However, we suspect that for the decentralized CBF-QP, since the agents do not know how other agents behave according to the current situation, the safety rate is low. Thus, we conclude that while centralized hand-crafted CBFs can maintain high safety rates, they cannot be used in practice for large-scale MAS due to their computational complexity. Decentralized CBFs can be applied in large-scale MAS, but the safety rate is lower than our approach because the centralized training of our approach gives the agents some consensus on their potential behaviors. In addition, our approach can deal with more complex dynamics where it is difficult to design hand-crafted CBFs.

## B.4 More Realistic Dynamics

One of the advantages of GCBF is that it is model-agnostic. To show that GCBF also works for other more realistic dynamics, we test GCBF for CrazyFlie dynamics. The 6-DOF quadrotor dynamics are given in [59] with $x \in \mathbb{R}^{12}$ consisting of positions, velocities, angular positions and angular

Table 4: Experimental results in the CrazyFlie environment

| Number of agents | Safety rate | Goal reaching rate | Success rate |
|---|---|---|---|
| 16 | $1.000 \pm 0.000$ | $0.996 \pm 0.015$ | $0.996 \pm 0.015$ |
| 32 | $1.000 \pm 0.000$ | $0.994 \pm 0.016$ | $0.994 \pm 0.016$ |
| 64 | $0.998 \pm 0.007$ | $0.997 \pm 0.008$ | $0.995 \pm 0.010$ |
| 128 | $0.993 \pm 0.010$ | $0.996 \pm 0.006$ | $0.989 \pm 0.012$ |

velocities, and $u \in \mathbb{R}^4$ consisting of the thrust at each of four motors :

$$\dot{p}_x = \big(c(\phi)c(\psi)s(\theta) + s(\phi)s(\psi)\big)w - \big(s(\psi)c(\phi) - c(\psi)s(\phi)s(\theta)\big)v + uc(\psi)c(\theta) \tag{12a}$$

$$\dot{p}_y = \big(s(\phi)s(\psi)s(\theta) + c(\phi)c(\psi)\big)v - \big(c(\psi)s(\phi) - s(\psi)c(\phi)s(\theta)\big)w + us(\psi)c(\theta) \tag{12b}$$

$$\dot{p}_z = w\,c(\psi)c(\phi) - u\,s(\theta) + v\,s(\phi)c(\theta) \tag{12c}$$

$$\dot{u} = r\,v - q\,w + g\,s(\theta) \tag{12d}$$

$$\dot{v} = p\,w - r\,u - g\,s(\phi)c(\theta) \tag{12e}$$

$$\dot{w} = q\,u - p\,v + \frac{U_1}{m} - g\,c(\theta)c(\phi) \tag{12f}$$

$$\dot{\phi} = r\frac{c(\phi)}{c(\theta)} + q\frac{s(\phi)}{c(\theta)} \tag{12g}$$

$$\dot{\theta} = q\,c(\phi) - r\,s(\phi) \tag{12h}$$

$$\dot{\psi} = p + r\,c(\phi)t(\theta) + q\,s(\phi)t(\theta) \tag{12i}$$

$$\dot{r} = \frac{1}{I_{zz}}\big(U_2 - pq(I_{yy} - I_{xx})\big) \tag{12j}$$

$$\dot{q} = \frac{1}{I_{yy}}\big(U_3 - pr(I_{xx} - I_{zz})\big) \tag{12k}$$

$$\dot{p} = \frac{1}{I_{xx}}\big(U_4 + qr(I_{zz} - I_{yy})\big) \tag{12l}$$

where $m, I_{xx}, I_{yy}, I_{zz}, k_r, k_t > 0$ are system parameters, $g = 9.8$ is the gravitational acceleration, $c(\cdot), s(\cdot), t(\cdot)$ denote $\cos(\cdot), \sin(\cdot), \tan(\cdot)$, respectively, $(p_x, p_y, p_z)$ denote the position of the quadrotor, $(\phi, \theta, \psi)$ its Euler angles and $u = (U_1, U_2, U_3, U_4)$ the input vector consisting of thrust $U_1$ and moments $U_2, U_3, U_4$.

The relation between the vector $u$ and the individual motor speeds is given as

$$\begin{bmatrix} U_1 \\ U_2 \\ U_3 \\ U_4 \end{bmatrix} = \begin{bmatrix} C_T & C_T & C_T & C_T \\ -dC_T\sqrt{2} & -dC_T\sqrt{2} & dC_T\sqrt{2} & dC_T\sqrt{2} \\ -dC_T\sqrt{2} & dC_T\sqrt{2} & dC_T\sqrt{2} & -dC_T\sqrt{2} \\ -C_D & C_D & -C_D & C_D \end{bmatrix} \begin{bmatrix} \omega_1^2 \\ \omega_2^2 \\ \omega_3^2 \\ \omega_4^2 \end{bmatrix}, \tag{13}$$

where $\omega_i$ is the angular speed of the $i-$th motor for $i \in \{1, 2, 3, 4\}$, $C_D$ is the drag coefficient and $C_T$ is the thrust coefficient. These parameters are given as: $I_{xx} = I_{yy} = 1.395 \times 10^{-5}$ kg-m$^2$, $I_{zz} = 2, 173 \times 10^{-5}$ kg-m$^2$, $m = 0.0299$ kg, $C_T = 3.1582 \times 10^{-10}$ N/rpm$^2$, $C_D = 7.9379 \times 10^{-12}$ N/rpm$^2$ and $d = 0.03973$ m (see [59]).

We use the same setting as the third set of environments introduced in Section 4, i.e., fix the density and the average traveling distance. The results are shown in Figure 10 and Table 4. We can observe that GCBF works well in the CrazyFlie dynamics, which supports the generalizability of GCBF in more complex realistic dynamics.

## C  Theoretical results

### C.1  Safety result: proof of Theorem 1

The proof of Theorem 1 is based on the CBF-based forward invariance arguments for time-varying safe sets [19, 60]. Since the edge features $e_{ij}$ depend on the states of the agent $i$ and its neighbor $j$,

the arguments of $h$ can be re-arranged to obtain the following form:

$$h(\bar{e}_i(t), \bar{v}_i(t)) = h\left(x_i(t), \left(x_{j_1}(t), x_{j_2}(t), \ldots, x_{j_{|\mathcal{N}_i|}}(t)\right), \bar{v}_i(t)\right), \tag{14}$$

which can be compactly written as $h(t, x_i)$ since the information of the neighbors is time-dependent. Now, the *total* time-derivative of $h$ reads

$$\dot{h}(t, x_i) = \frac{\partial}{\partial t} h(t, x) + \frac{\partial}{\partial x_i} h(t, x_i) F(x_i, u_i). \tag{15}$$

Now, from [60, Theorem 1], it holds that the set $S(t) = \{x_i \; h(t, x_i) \geq 0\}$ is forward invariant for the unique solutions of the dynamics $\dot{x}_i = F(x_i, u_i)$ if $h$ is a valid CBF, i.e., if there exists a class-$\mathcal{K}$ function $\alpha$ such that $\sup u \dot{h}(t, x_i) \geq -\alpha(h(t, x_i))$. Now, per Theorem 1, let $u_i$ be a smooth control input chosen from the set $\mathcal{U}_i^{safe}$. Then, it holds that $\frac{\partial}{\partial t} h(t, x) + \frac{\partial}{\partial x_i} h(t, x_i) \dot{x}_i \geq -\alpha(h(t, x_i))$, and thus, $h$ is a valid CBF. Finally, since $u_i$ is smooth and the dynamical function $F$ is assumed to be locally Lipschitz continuous, it holds that the closed-loop dynamics $F(\cdot, u_i(\cdot, t))$ is locally Lipschitz continuous, and hence, the closed-loop solutions exist and are uniquely determined. Thus, it can be concluded that the set $S(t)$ is forward invariant for the closed-loop system under the conditions of Theorem 1, which completes the proof.

## C.2 Generalization result

**Correctness of the learned control policy** The generalization results for the learned control policy for safety objectives dictate that the learned controller can keep the system safe even under unseen scenarios with high probability. The generalization results in [20, Proposition 3] states that generalization error remains bounded with high probability, and under certain regularity assumptions (such as Lipschitz continuity) or functions drawn from Reproducing Kernel Hilbert Space, this error vanishes with the increase in the number of samples. The same generalization error-bound result holds for our learned controller and for the sake of brevity, we refer interested readers to [20].

