# OpenReview forum: "Neural Graph Control Barrier Functions Guided Distributed Collision-avoidance Multi-agent Control"
_robot-learning.org/CoRL/2023/Conference — CoRL 2023 Poster_

### Official Review · Reviewer_pBaq · 2023-06-27

**Confidence:** 4
**Originality:** Very Good
**Technical Quality:** Very Good
**Clarity Of Presentation:** Excellent
**Impact:** 3

**Recommendation:**

Weak Accept: I recommend accepting the paper, but will not argue for my recommendation if the majority of other reviewers have a different opinion.

**Review:**

### Strengths

The paper demonstrates several notable strengths:

- Synergy of control and machine learning: The integration of control barrier functions (CBFs) with graph neural networks (GNNs) combines control theory with machine learning. This combination is both intriguing and promising for safe multi-agent control.
- Relevance of the topic: The problem of distributed safe multi-agent control in large-scale environments with moving obstacles is highly relevant in the context of multi-agent systems and has applications in a plethora of real-world problems.
- Clear and well-written: The paper stands out for its clarity and readability. The inclusion of clear figures and comprehensive descriptions aids in understanding the proposed approach and the experimental results.
- Code availability and experimental details: The authors provde code and extensive experimental details in the supplementary material. This transparency enables reproducibility and facilitates further exploration and validation of the proposed method.
- Convincing experiments against strong baselines: The paper presents persuasive experimental results by comparing the proposed approach against strong baseline methods like MAPPO and multi-agent distributed CBF (MDCBF). This empirical evidence showcases the superior performance and effectiveness of the presented approach in a variety of scenarios.

Overall, these strengths contribute to the significance and impact of the paper, establishing its merit within the field of multi-agent systems and machine learning.

### Weaknesses

The weaknesses of the approach are mostly and adequately addressed in the limitations section of the paper. Beyond that, some additional concerns are listed below

- Potential sensitivity to hyperparameters: The paper lacks sufficient justification for the choice of hyperparameters, such as the weights of the losses in the Appendix and the choice of the function $\alpha(h(x))$. Conducting experiments with different parameter choices or explaining the choices that are made would provide insights into the method's behavior and robustness.
- Inconsistent experimental setups: The third experiment, which involves obstacles, differs significantly from the first two experiments. Additionally, the third experiment does not increase the number of agents during inference. Providing a more consistent experimental setup and explaining the rationale behind these differences would improve the clarity of the results.
- The connection between the fixed-size inputs for the mathematical definition on Page 4, and the input to the GNN could be stated more clearly. While this is briefly explained in Footnote 3, it would be appreciated if the authors could clearly distinguish between theory and practical application, and provide a brief explanation for how the two differ.
- Additional visualizations: While videos are available in the supplementary material, incorporating additional "strips" of policies at critical timesteps in the main paper or appendix would enhance the visualization of the approach. Furthermore, including a visualization for the 3D drone task, such as e.g., a 2D image with depth colorization, would be insightful.
- Typos:
    - L. 275: “Detach” → “Detect”
    - L. 94 “assumed to” → “assumed to be”

**Quality Of The Limitations Section:**

Limitations are addressed clearly

**Questions For Rebuttal:**

- How exactly is the online policy refinement performed, and how does this affect e.g., the runtime of the algorithm?
- How does the experiment with obstacles behave when the number of agents is increased during inference?
- Some of the parts of the Appendix, e.g., the ablations in Appendix B are not referenced in the main paper. Referencing them would strengthen the claims of the main paper and make it more coherent.

**Robotics Focus:**

Highly relevant to robotics but no hardware experiments

**Summary Of Paper:**

The paper presents a novel approach called graph control barrier functions (GCBFs) for addressing the challenges of distributed safe multi-agent control in large-scale environments with moving obstacles. The method combines control barrier functions for safe with a graph structure for scalable and generalizable decentralized control. The authors utilize graph neural networks to learn both the neural GCBF and the distributed control policy. They further extend the framework to handle LiDAR-based point cloud inputs, allowing for obstacle detection and thus making it more applicable to real-world robotics settings. The empirical results demonstrate the superiority of the presented approach over existing methods on $2$d navigation tasks, achieving significantly higher safety and goal-reaching rates when increasing numbers of agents during inference. The proposed approach is completely distributed and scalable, making it suitable for large-scale multi-agent systems.

**Summary Of Recommendation:**

The paper is enjoyable to read, demonstrates a strong foundation, and has a convincing experimental section. The availability of code and comprehensive experimental details in the supplementary material further contribute to its strength. A large part of the weaknesses identified can be addressed through revisions and clarifications during the rebuttal and revision process. Unless any major issues are discovered during the rebuttal, I recommend accepting the paper.

Edit: I want to thank the reviewers for the detailed discussion and for clearing up a majority of my concerns. I will remain with my original recommendation for a weak accept, but am now more certain of the contributions of the paper and their significance for CoRL. I raised the confidence from a 3 to a 4 to represent this.

---

### Official Review · Reviewer_dLpN · 2023-07-16

**Confidence:** 4
**Originality:** Fair
**Technical Quality:** Very Good
**Clarity Of Presentation:** Very Good
**Impact:** 3

**Recommendation:**

Weak Accept: I recommend accepting the paper, but will not argue for my recommendation if the majority of other reviewers have a different opinion.

**Review:**

Strengths:

Overall the paper is well-written and well-motivated. The provided code seems to implement exactly the proposed method.

Weaknesses:

1- The contribution seems incremental wrt to the reference [20] of the paper. The main difference wrt [20] is that this article proposes using the GNN to learn the safe action policy and use the learned control barrier function as a switching policy between the nominal and learned safe controller. I would recommend the authors revise their claimed contributions.

2 - There are only simulation results. Adding real robot experiments would increase this paper's contribution.

**Quality Of The Limitations Section:**

Limitations are addressed clearly

**Questions For Rebuttal:**

1 - Can you please clearly state the contribution of this article wrt [20]?

2 - Liveness seems to be not well defined. Would a better definition of liveness be that there is a t such that ||p_i(t)-p_i^{goal}|| < \epsilon, where \epsilon is a distance threshold? \epsilon can be the same as r. Can the authors explain the proposed definition?

3 - How does the proposed approach compare performance with an optimization-based control barrier function? It would be good to present results for a baseline following one of the approaches as proposed in reference [15] of the paper.

4 - In the training loss, there is only one objective the affect the safe policy weights, which is the normal distance to the nominal control action. Can the authors explain how such an objective ensures that we are learning a safe policy action?

**Robotics Focus:**

Relevant but unlikely to deploy to hardware in near future

**Summary Of Paper:**

This paper proposes to use Graph Neural Networks (GNNs) to learn a safety score and safe action policy. The main idea is to formulate the safety score as a control barrier function and use it to switch between a nominal and a learned-based controller.

**Summary Of Recommendation:**

The paper is well-written and technically sound. The major weakness is the contribution which seems to be minor and incremental wrt to reference [20] in the paper. Moreover, there are no real robotics experiments that could be a major contribution wrt the literature.

---

### Official Review · Reviewer_fBfM · 2023-07-18

**Confidence:** 2
**Originality:** Good
**Technical Quality:** Good
**Clarity Of Presentation:** Very Good
**Impact:** 3

**Recommendation:**

Weak Accept: I recommend accepting the paper, but will not argue for my recommendation if the majority of other reviewers have a different opinion.

**Review:**

### Strengths
S1) Interesting deep learning approach combining multiple different methods like CBFs, GNNs and attention mechanism.

S2) Highly scalable.

S3) Training can be done in a relatively short amount of time (only 1 hour).

### Weaknesses
W1) The paper states that its controller is safe, although it has no safety guarantees (as stated by the paper itself in the limitations).

W2) While the paper derives a new type of CBFs called Graph CBF, the theoretical analysis is incomplete. The proof for the given theorem (if the policy satisfies CBF properties, then there are no collisions) is skipped and there is no further analysis (like existence results).

W3) It is not clear how the method in this paper compares to classical non-learning methods.

W4) The paper mentions a policy refinement strategy in their method, but does not investigate it further in the experiments section.

### Evaluation
The proposed learning method is interesting and the experimental results are impressive.
However, the paper uses the word safe in title, abstract and main text although the proposed method is not inherently safe, which the paper itself states in the limitation section.
The wording in the paper should be changed to account for this.
It is also not clear how the method in this paper compares to classical non-learning methods, which should be addressed.

**Quality Of The Limitations Section:**

Limitations are addressed clearly

**Questions For Rebuttal:**

1. The wording in title, abstract and main body of the paper refers to the learned controller as being safe. This is slightly missleading, as the controller has no theoretical guarantees. While it is motivated out of CBFs there are no guarantees given weather there exists a CBF and weather the training converges to it in final time. Maybe rename the "safe controller" to "collision avoidance controller" or something similar.
2. Formula 5 is not clear to me. Where does the sum come from? Is it a batch (the same agent simulated multiple times in different settings at once)?.
3. Page 6 describes two different ways to calculate \dot{h}. (Lines 207/208 and the paragraph beginning with Line 212). Please clarify this.
4. In the appendix the training time is given (1 hour on described computer). Please state this in the main text and also give the training times of MDCBF and MAPPO too. For me just 1 hour training time sound very short so this could also be an additional contribution.
5. In Line 295 it is said that the success rate is very close to 1. Please use exact numbers.
6. The experimental section compares the algorithm to two other learning based methods. It would be also interesting to see how it compares to classic methods like potential fields to see the benefits of learning.
7. A short section in the video describing the overall method and approach could be added.
8. In the video, it takes a long time for a few agents in the experiments to reach their goal. Maybe show the beginning in normal speed and once the majority of agents have reached their goal speed up the video.
9. How often does the policy refinement strategy have to be used in the experiments?

**Robotics Focus:**

Highly relevant to robotics but no hardware experiments

**Summary Of Paper:**

The paper proposes to train a graph Neural Network (GNN) to control mobile robots. Every robot is equipped with a GNN and receives information from its neighbour environment, making the method distributed. A loss function based on Control Barrier Functions (CBF) is used to avoid inter-agent collisions between the robots and collisions of the robots with static obstacles. The GNN formulation makes the proposed method scalable with the size of the swarm and the paper shows that with its method more than 1000 agents can be controlled although the network was trained on only 16 agents.

**Summary Of Recommendation:**

While the paper is written well and the method is applicable for huge swarms, its methods lacks of a theoretical analysis. The statements in the paper about the method being safe should be reformulated to account for this. Furthermore, see main strengths and weaknesses as listed above.

---

### Official Review · Reviewer_umVL · 2023-07-20

**Confidence:** 5
**Originality:** Good
**Technical Quality:** Good
**Clarity Of Presentation:** Very Good
**Impact:** 3

**Recommendation:**

Weak Accept: I recommend accepting the paper, but will not argue for my recommendation if the majority of other reviewers have a different opinion.

**Review:**

The paper addresses a relevant problem in multirobot control and proposes an interesting method with graph neural networks to achieve it within the control barrier function framework. The paper shows a significant increase over two recent methods with a very large number of robots. Also, the limitations section discusses reasonably the main points observed during the experiments, identifying potential future work.

There are some directions of improvement for the paper:
- a popular type of approach for multi-robot collision avoidance is "velocity obstacle", which should be included in the discussion of the related work at least, and as part of the comparison. It would be also interesting to have a comparison with an optimal planning strategy.

- the experiments include idealized kinematic models; it would be good to use a realistic simulator or enrich the current model with noise.

- the statement that planning approaches "do not generalize to new unseen environments." is not clear, as those approaches, given the map of an environment, will calculate the corresponding path -- unseen is typically interpreted as not seen perhaps during the experiments. This point should be clarified and rephrased.

- the appendix could include the proof of Theorem 1.

A couple of typos: derivates -> derivatives; "LiDAR to detach obstacles" -> "... detect ..."

**Quality Of The Limitations Section:**

Limitations are addressed clearly

**Questions For Rebuttal:**

No specific questions apart from the suggestions for improvement.

**Robotics Focus:**

Highly relevant to robotics but no hardware experiments

**Summary Of Paper:**

The paper addresses the problem of controlling robots to get to their destination without collision, by proposing graph control barrier functions to encode safety constraints and a graph neural network to learn the corresponding control. Experiments are performed in simulation using different kinematic models for the robot, comparing with other two approaches.

**Summary Of Recommendation:**

While the paper could be improved on the experimental side, the paper overall shows an interesting method based on GCBF and GNNs that significantly improves the performance in scenarios with a large number of robots. Please see the details in the "Review" box.

---

### Official Review · Reviewer_QaEJ · 2023-07-20

**Confidence:** 4
**Originality:** Fair
**Technical Quality:** Fair
**Clarity Of Presentation:** Very Good
**Impact:** 2

**Recommendation:**

Weak Reject: I recommend rejecting the paper, but will not argue for my recommendation if the majority of other reviewers have a different opinion.

**Review:**

- The paper comes across as a heuristic for safety. It builds a false notion of safety without guaranteeing any actual safety. Calling the trained neural network a Control Barrier Function is misleading since the trained network is not guaranteed to satisfy the CBF conditions (presented in equation (1)).

- The loss function depends on the empirical evaluation around the data points used for training. This raises the question of the effect of the data quality used during training on the correctness of the proposed framework.

- Which assumptions on the robot dynamics/workspace are needed to ensure the safety of the multi-agent system? When can one trust the decisions taken by GCBF to be safe?

- The numerical results section is inconclusive. In particular, the paper lacks to compare against model-based (manually designed) CBF-based methods. The dynamics of the 2D vehicle used in simulation have been widely studied, and manually designed CBFs for multi-agents of these robots have been presented in the literature.

- The limitations section claims that it is hard, if not impossible, to ensure that the proposed method ensures the system's safety. First, this claim was not supported by any evidence. Second, if the claim is true, this points out that the proposed approach may not be the right one to handle a crucial and consequential issue like safety.

**Quality Of The Limitations Section:**

Additional details required

**Questions For Rebuttal:**

(1) What is the effect of the data quality used during training on the correctness of the proposed framework?

(2) Which assumptions on the robot dynamics/workspace are needed to ensure the safety of the multi-agent system? When can one trust the decisions taken by GCBF to be safe?

(3) How does GCBF perform against manually designed CBFs for 2D ground vehicles?

**Robotics Focus:**

Relevant but unlikely to deploy to hardware in near future

**Summary Of Paper:**

This paper proposes a safety mechanism for multi-agent systems using Control Barrier Functions (CBF). The proposed framework trains a graph neural network to learn both the controller and a barrier certificate. The paper provides numerical simulations to assess the scalability of the proposed framework using 2D ground vehicles and 3D drones.

**Summary Of Recommendation:**

In its current format, the paper proposes a heuristic to promote safe control actions during training. The paper does not provide any clear assumptions under which the proposed framework is guaranteed to provide safe control actions. Moreover, the numerical results lack comparison against the model-based design of CBFs. Given the criticality of safety, I do not recommend this paper for inclusion in CoRL 2023.

---

### Author Response · Authors · 2023-08-10
**General Remark (part 1)**

We thank all the reviewers for their time and efforts in evaluating our manuscript and giving constructive remarks and suggestions. We are delighted that the reviewers find our paper interesting (fBfM, pBaq), having very good experimental results (umVL, fBfM, pBaq), and well-written (all the reviewers). We respond to each of the questions by the reviewers in detail. Here, we summarize the main criticism and how we have addressed them. We have also uploaded a revised version of our paper.

1. **Comparison with handcrafted CBF-QPs**
We conducted further experiments with a handcrafted CBF for the SimpleCar experiments as it is relatively easier to design a handcrafted CBF for a double-integrator system. We designed two frameworks:

- Centralized CBF: In this framework, inputs of all the agents are solved together by setting up a *centralized* QP containing CBF constraints of all the agents.
- Decentralized CBF: In this framework, each agent computes its own input through a QP with CBF constraint for its own safety and assumes the neighbors take the nominal control $\pi_\mathrm{nom}$ as it is not possible to obtain the current control action in a synchronous setup.

    We show the *safety rate* results in the table below:

    |Number of agents|Centralized|Decentralized|
    |---|---|---|
    |16|1.0|0.945|
    |32|1.0|0.820|
    |64|1.0|0.734|
    |128|1.0|0.550|

    We also show the *computational time (seconds)* per control step in the table below:

    |Number of agents|Centralized|Decentralized|
    |---|---|---|
    |16|0.009|0.00047|
    |32|0.013|0.00064|
    |64|0.039|0.00096|
    |128|0.244|0.00156|

    Our experiments illustrated that the centralized hand-crafted CBF-QP-based controller can maintain a safety rate of 1.0. However, it is computationally very costly to solve CBF-QP for a large-scale MAS. In particular, we found out that for up to 64 agents, the CBF-QP can be used for "real-time" control synthesis (discrete-time step of 0.03s), but for more agents, the CBF-QP is too slow (0.244s per step for 128 agents) and cannot be used for "real-time" control synthesis. We also tried to increase the control time step to 0.25s for 128 agents so that the CBF-QP is "real-time", but the safety rate drops to 0.55.

    The decentralized QP, on the other hand, is fast enough to be used for real-time control synthesis for a large number of agents. However, we suspect that for the decentralized CBF-QP, since the agents do not have knowledge of how other agents behave according to the current situation, the safety rate is low.

    Thus, our conclusion is that while centralized hand-crafted CBFs can maintain high safety rates, they cannot be used in practice for large-scale MAS due to their computational complexity. Decentralized CBFs can be applied in large-scale MAS, but the safety rate is lower than our approach, because the centralized training of our approach makes the agents have some consensus on their potential behaviors. In addition, our approach can deal with more complex dynamics where it is difficult to design hand-crafted CBFs.

    We have also added these discussions in Appendix B.3 in the revised manuscript.

---

> ### Author Response · Authors · 2023-08-10
> **General Remark (part 2)**
>
> 2. **Notion of safety in the paper and trusting GCBF decision**
>
>     Based on the suggestions and questions raised by the reviewer (QaEJ and fBfM), we have modified the presentation of the manuscript so that we do not claim that the proposed framework results in a CBF, and that it can guarantee safety. The proposed framework is a CBF-guided approach, and like many previous neural certificate approaches, authors commonly used "safety" in the context of a learned barrier certificate. However, we admit that this is not very rigorous and we made the modifications in the revised manuscript to reflect these changes. The main changes are as follows:
>     - We changed the title from "Distributed Safe Multi-agent Control Using Neural Graph Control Barrier Functions" to "Neural Graph Control Barrier Functions Guided Distributed Collision-avoidance Multi-agent Control"
>     - We changed the notion of "safe" controller/constraint to "collision-avoidance" controller/constraint (in the abstract, in the list of contributions as well as the main text throughout the paper)
>     - We changed the learned controller from $u^\mathrm{safe}$ to $u^\mathrm{NN}$ in Figure 1 as well the corresponding text on Page 5.
>
>     Furthermore, we would like to clarify that for NN-based certificates, it is very difficult to formally verify that the learned function will satisfy the required conditions (in this case, the three conditions of Eq. (2)). In theory, verification of correctness of an NN-based certificate can be carried out using tools such as SMT-solvers. However, such verification problems are computationally intractable for deep NNs and complex NN architectures, such as a GNN. Formal verification of NN is an NP-complete problem, even for simple NN architectures. There is very little work on the verification of GNNs, one such example is:
>
>     *Wu, Haoze, et al. "Scalable verification of GNN-based job schedulers." Proceedings of the ACM on Programming Languages 6.OOPSLA2 (2022): 1036-1065.*
>
>     However, even the method from the aforementioned paper takes many hours and engineering nuts and bolts to verify GNN with 2 hidden layers and 32 and 16 hidden units on each layer, since verifying GNN needs unrolling the system. It is very unlikely that there are tools or solvers available to handle the GCBF-like large GNN architecture that consists of 4 MLP layers with sizes 2048x2048, 128x128, 2048x2048, and 512x128x32.
>
>     Another difficulty for us to verify the NN architecture used for GCBF is caused by the LiDAR inputs. The LiDAR input represents different possibilities of obstacles, and it is impossible to bound all possible obstacle distributions. Most NN verification tools work for bounded input and since the input to the NN in our case is unbounded, the existing tools cannot be directly used for verification of the proposed GCBF architecture.
>
>     Our numerical experiments empirically verify the correctness of the learned controller. We can provide additional empirical evidence on the correctness of the learned GCBF. While the test results illustrate that the learned GCBF can maintain a high safety rate for various scenarios across various environments, we report the accuracy of satisfaction of the three conditions in Eq. (2) by the learned GCBF in the Dubins Car environment with 16 agents here:
>
>     - $\dot{h}\geq\alpha(h)$ in safe set: 0.9960;
>     -  $h>0$ in safe set: 0.9940,
>     -  $h<0$ in unsafe set: 0.9752.
>     The first two conditions are evaluated using 500 simulation trajectories. For the unsafe condition, however, since we do not have unsafe states in the tests (the safety rate is 1.0 here), we evaluate them on 48000 random sampled unsafe states.

---

> > ### Author Response · Authors · 2023-08-10
> > **General Remark (part 3)**
> >
> > 3. **Policy iteration and its effect on time complexity**
> >
> >     We performed ablation studies on the *max-iter* parameter in the online policy refinement to see how it affects the performance as well the computational time of the algorithm. The results are given in the following table for 64 agents tested in the DubinsCar environment.
> >
> >     | max iter | success rate |  per step per agent  computation time (s) |
> >     |----------|--------------|--------------------------------------|
> >     |     0    | 0.977 |                              0.0004 |
> >     |    30    |  0.978 |                            0.0013 |
> >     |    60    | 0.979 |                             0.0026 |
> >     |    90    | 0.981 |                              0.0036 |
> >
> >     As expected, the per-step computation time increases with an increase in the *max-iter* parameter, and so does the success rate as the policy refinement helps in getting a higher safety rate. Furthermore, it can be observed that the per-agent computational time is still very low, even for 90 max-iterations of online policy refinement. Thus, it can be concluded that the policy refinement can be used without compromising on the real-time nature of the control synthesis, since the GCBF-control framework is a decentralized control framework.
> >     We have also included these results along with a plot on *safety-rate* vs *max-iter* in the revised manuscript in Appendix B.2
> >
> > 4.  **Experiments with obstacles when the number of agents is increased**
> >
> >     Per the suggestions of reviewer pBaq, we conducted additional experiments with an increasing number of agents in the obstacle environment. In this case, we increased the number of obstacles along with the number of agents, so that the ratio of the number of agents (N) to the number of obstacles $N_{obs}$ remains constant ($N / N_{obs} = 2$ and $4$). The results are tabulated below.
> >
> >     | $N$   | $N_{obs}$ | $N / N_{obs}$ |  safety rate | reaching rate | success rate |
> >     |-----|-----------|:-------------:|:------------:|:-------------:|:------------:|
> >     |  16 |         4 |             4 |            1 |             1 |            1 |
> >     |  32 |         8 |             4 |            1 |             1 |            1 |
> >     |  64 |        16 |             4 |  0.998046875 |  0.9951171875 | 0.9931640625 |
> >     | 128 |        32 |             4 | 0.9912109375 |  0.8447265625 | 0.8413085938 |
> >     |  16 |         8 |             2 |            1 |             1 |            1 |
> >     |  32 |        16 |             2 |  0.994140625 |   0.998046875 |    0.9921875 |
> >     |  64 |        32 |             2 | 0.9970703125 |  0.9814453125 |  0.978515625 |
> >
> >     The data demonstrates that the GCBF maintains a high safety rate in the case with larger number of agents and higher number of obstacles as well. While the safety rate with 128 agents is still close to 1, the reaching rate drops in this case due to the environment getting overcrowded with so many obstacles.
> >
> >     We have included the results for the case when $N / N_{obs} = 4$ in Appendix B.1 in the revised manuscript.

---

### Decision · Program_Chairs · 2023-08-30

**Decision:**

Accept (Poster)

**Comment:**

The paper proposes a method for distributed control of multiple agents with graph barrier functions encoding safety constraints and a graph neural network to learn the controller. Experiments were performed in simulation for both 2D and 3D cases.

The approach is novel and connects geometry deep learning with multiple methods such as CBFs, GNNs and attention mechanisms. It is also highly scalable and fast to train.

Reviewers were mostly positive about the paper but there were also a few concerns, for example:
- Calling the trained neural network a Control Barrier Function is misleading since the trained network is not guaranteed to satisfy the CBF conditions (presented in equation (1)); which the authors agree with.
-  Comparison against model-based (manually designed) CBF-based methods.
- The policy refinement strategy in the method is not discussed in the experiments section.

The authors have addressed these and other issues adequately in the rebuttal and provided an improved version of the paper with a different title, clarifying the most critical issue.